# Soil elemental changes during human decomposition

**Lois S. Taylor**[1]*, **Adrian Gonzalez**[2], **Michael E. Essington**[1], **Scott C. Lenaghan**[3], **C. Neal Stewart**[4], **Amy Z. Mundorff**[5], **Dawnie W. Steadman**[5], **Jennifer M. DeBruyn**[1]*

**1** Department of Biosystems Engineering & Soil Science, University of Tennessee, Knoxville, TN, United States of America, **2** Department of Civil and Environmental Engineering, University of Tennessee, Knoxville, TN, United States of America, **3** Center for Agricultural Synthetic Biology, University of Tennessee, Knoxville, TN, United States of America, **4** Department of Plant Sciences, University of Tennessee, Knoxville, TN, United States of America, **5** Department of Anthropology, University of Tennessee, Knoxville, TN, United States of America

* ltaylo32@vols.utk.edu (LST); jdebruyn@utk.edu (JMD)

**Data Availability Statement:** Data can be found as a dataset in the Supporting Information files and in the github repository: https://github.com/jdebruyn/SPOT-soilelements.

## Abstract

Mammalian decomposition provides pulses of organic matter to the local ecosystem creating ephemeral hotspots of nutrient cycling. While changes to soil biogeochemistry in these hotspots have been described for C and N, patterns associated with deposition and cycling of other elements have not received the same attention. The goal of our study was to evaluate temporal changes to a broad suite of dissolved elements in soils impacted by human decomposition on the soil surface including: 1) abundant mineral elements in the human body (K, Na, S, P, Ca, and Mg), 2) trace elements in the human body (Fe, Mn, Se, Zn, Cu, Co, and B), and 3) Al which is transient in the human body but common in soils. We performed a four-month human decomposition trial at the University of Tennessee Anthropology Research Facility and quantified elemental concentrations dissolved in the soil solution, targeting the mobile and bioavailable fraction. We identified three groups of elements based on their temporal patterns. Group 1 elements appeared to be cadaver-derived (Na, K, P, S) and their persistence in soil varied based upon soluble organic forms (P), the dynamics of the soil exchange complex (Na, K), and gradual releases attributable to microbial degradation (S). Group 2 elements (Ca, Mg, Mn, Se, B) included three elements that have greater concentrations in soil than would be expected based on cadaver inputs alone, suggesting that these elements partially originate from the soil exchange (Ca, Mg), or are solubilized as a result of soil acidification (Mn). Group 3 elements (Fe, Cu, Zn, Co, Al) increased late in the decomposition process, suggesting a gradual solubilization from soil minerals under acidic pH conditions. This work presents a detailed longitudinal characterization of changes in dissolved soil elements during human decomposition furthering our understanding of elemental deposition and cycling in these environments.

## Introduction

Mammalian decomposition is a dynamic process that provides pulses of organic matter to the local ecosystem. These nutrient pulses are dispersed throughout the wider ecosystem in a

**Funding:** The authors acknowledge funding by the Defense Advanced Research Projects Agency (Award No. D20AC00007) to CNS, SCL, AZM, DWS and JMD. The views, opinions, and/or findings expressed are those of the authors and should not be interpreted as representing the official views of policies of the Department of Defense or the United States Government (Approved for Public Release, Distribution Unlimited). Publication fees were granted by the University of Tennessee Librarians Open Publishing Support Fund. The funders had no role in study design, data collection and analysis, decision to publish, or preparation of the manuscript.

**Competing interests:** The authors have declared no competing interests exist.

variety of ways: into the atmosphere via gaseous emissions (including volatile organic compounds, or VOCs) [1], into insects and scavengers which may move materials over large distances [2–4], and into the soil creating ephemeral hotspots of enhanced nutrient cycling and microbial activity [5–8]. Changes to soil biogeochemistry in these hotspots have been described, including pH [5–20], electrical conductivity (EC) [6,7,10,12,13,15,17,20,21], organic and inorganic carbon (C) and nitrogen (N) speciation and cycling [5–15,17–19,21–28], and soil oxygen [7]. However, patterns associated with the deposition and cycling of elements other than C and N have not been explored with the same degree of detail.

The most abundant elements in the human body are oxygen (O), carbon (C), hydrogen (H), nitrogen (N), calcium (Ca), phosphorus (P), potassium (K), sulfur (S), sodium (Na), chloride (the mineral ion of chlorine), and magnesium (Mg), together comprising greater than 99% of total body mass [29]. Based on elemental proportions, it is estimated that a 70 kg human is composed of between 1 and 38.6 kg each of O, C, H, and N, and that the remaining mineral elements are found in abundances of less than 1 kg each, all of which could theoretically be transferred to the soil assuming no losses to atmosphere, insect activity, or scavenging. H and O are distributed throughout decomposition fluid, VOCs, and water. Of the remaining elements, total C and N, their organic and inorganic fractions, and their transformation products are the most commonly documented for surface decomposition [5,7–15,17–19,21–28]. This is followed by the plant nutrients P [5,10,12,13,15–19,21,22,24–26,28] and K [9,10,16–18,22,25,28,30], the latter of which is also an exchangeable soil cation. In contrast, soil changes of other exchangeable cations, Ca, Na, and Mg, have received only general characterization [9,10,16,17,25,26,28,30], and while the presence of S and chloride have been noted, their soil concentration changes are poorly described [9,10,25]. To our knowledge other cadaver-derived elements have received no attention. Currently, the most detailed longitudinal characterizations of elemental pools have been performed using pigs rather than humans [16,18,19,21]. While frequently used as human surrogates, there are differences between pig and human tissue composition [31], decomposition patterns [8,32,33], and soil chemistry responses, particularly with respect to pH [8,24] (but see [5,8–11,14,16–20]). Furthermore, the collective surface decomposition literature spans a diversity of soil types, each having unique chemistry, and therefore complicating interpretation of ultimate availability of resource pools for plant uptake and microbial cycling. Our study was motivated by the lack of information regarding how elemental pools change during human decomposition, in order to provide insight into macro- and micronutrient cycling during decomposition.

The goal of our study was to evaluate temporal changes to a broad suite of elements in soil solutions impacted by surface decomposition of whole human remains: 1) abundant mineral elements in the human body (K, Na, S, P, Ca, and Mg), 2) elements found in trace abundances in the human body (Fe, Mn, Se, Zn, Cu, Co, and B), and 3) Al which is largely transient in the human body but common in soils [34]. The greatest and least abundant of these commonly occurring mineral elements in a 70 kg living human body could ultimately provide up to 1 kg (Ca) and 14 g (Mg), respectively, to the soil during decomposition, assuming no losses. Based upon this potentially substantial soil augmentation, we hypothesized that we would observe significant increases in these elements in decomposition-impacted soils relative to control soils unimpacted by decomposition, and that these changes would yield different temporal patterns. During tissue decomposition monovalent cations are lost first, followed by N and S derived from soft tissues, and finally P, Mg, and Ca derived from nucleic acids and recalcitrant tissues including bone [25]. Some of these elements derived from the breakdown of tissues are also exchangeable base cations in soil (i.e., $K^+$, $Na^+$, $Mg^{2+}$, and $Ca^{2+}$) and others have pH-dependent solubility in soil (i.e., Fe, Mn, Se, Zn, Cu, Co, and B), indicating that changes in the soil chemical environment during decomposition may affect dissolved pools. To test our

hypotheses, we performed a four-month human decomposition trial at the University of Tennessee Anthropology Research Facility and quantified elemental concentrations dissolved in the soil solution, with the intention of targeting the mobile (exchangeable) and bioavailable fraction. This work presents a detailed longitudinal characterization of changes in soil elemental profiles during human decomposition furthering our understanding of nutrient deposition and cycling in these environments. Assessing time-resolved changes to multiple elemental pools may have utility both ecologically and forensically by identifying biogeochemical cycling pathways other than C and N as well as potentially identifying markers of decomposition progression leading to better estimates of the postmortem interval (PMI). The soil solution phase is easily extracted and analyzed relative to other soil components, potentially making it a practical option for forensic use.

## Materials and methods

### Experimental site

This study was conducted at the University of Tennessee Anthropology Research Facility (ARF), Knoxville, TN, United States (35˚ 56' 28" N, 83˚ 56' 25" W). The ARF is part of the University of Tennessee Forensic Anthropology Center (FAC) and was the first established outdoor human decomposition laboratory [35]. The facility is situated on a ridgeline overlooking the Tennessee River and consists of temperate mixed deciduous forest with a Köppen climate classification of *Cfa* (humid subtropical). Soils are a mixture of Loyston-Talbott-Rock outcrop (LtD) and Coghill-Corryton (CcD) complexes and are classified as Lithic and Typic Hapludalfs and Typic Hapludults that are broadly composed of clayey residuum derived from weathered limestone. Soil profiles typically consist of silty clay loam and channery clay that extend to a lithic bedrock. These soils are frequently found in Southern Appalachian Mountain ridges and slopes [36,37].

### Site layout and donor placement

Three deceased human subjects (donors) were used in this study, which were donations to the FAC (https://fac.utk.edu/body-donation/) specifically for the purposes of decomposition research. No living human subjects were involved, and therefore this study was exempt from review by the University of Tennessee Institutional Review Board. No preferences were made for sex, ancestry, age, or height. Donors had not been autopsied nor were observed to have sustained physical trauma that might create artificial points of ingress for insects, microbes, or scavengers. As per all research at the Anthropology Research Facility, a research request was submitted to the FAC that detailed the research question, scope, methods and duration of work at the Facility. The request was reviewed and approved by the Director and a panel of senior staff of the FAC. The FAC does not issue formal permits beyond the written approval of the work to commence.

One female and two male donors (n = 3) within a weight range of 69.9 to 93.9 kg were accepted for this study (1). Donors were immediately frozen after intake to allow for simultaneous placement at the field site. Donors were placed at the study site unclothed, supine, and in direct contact with the soil surface to allow access to insects and scavengers, and were spaced across the facility. Data were collected from 9 April 2021 through 9 August 2021 (122 days) and consisted of local environmental data and soil samples for soil chemistry.

Local ambient temperature data were collected from the Knoxville Municipal Tyson McGhee Airport (KTYS) Automated Surface Observation System (ASOS). Accumulated degree hours (ADH) were determined by cumulative sums of hourly temperature data. Accumulated degree days Degree Days (ADD) were determined by cumulative sums of daily temperature means. TYS is located at a similar elevation and 16 km south of the study site.

**Table 1. Human donor data.**

| Donor ID | Sex | Age | Weight (kg) | Cause of death |
|---|---|---|---|---|
| D3_SPOT1_J | Male | 66 | 73.0 | Natural |
| D3_SPOT2_H | Male | 62 | 93.9 | Natural |
| D3_SPOT3_F | Female | 69 | 69.9 | Natural |

Data table for donors enrolled in the study. Donor study ID, sex, age, weight (kg), and cause of death are shown.

## Sample collection

Soil samples were taken at 27 time points throughout the decomposition process (Table 2). Pending weather, samples were taken two to three times per week for the first two months, weekly for three weeks, and every two weeks thereafter. Soil sampling was performed using a 1.9-cm diameter soil auger to remove five 15-cm cores from impacted soils immediately

**Table 2. Donor sampling time points.**

| Study Day | Date (D-M-Y) | ADH ambient | ADD |
|---|---|---|---|
| 0 | 9 April 2021 | 0 | 0 |
| 3 | 12 April 2021 | 1305 | 54.4 |
| 5 | 14 April 2021 | 2143 | 89.3 |
| 7 | 16 April 2021 | 2805 | 116.9 |
| 10 | 19 April 2021 | 3757 | 156.5 |
| 14 | 23 April 2021 | 4751 | 198 |
| 17 | 26 April 2021 | 5577 | 232.4 |
| 19 | 28 April 2021 | 6570 | 273.7 |
| 21 | 30 April 2021 | 7586 | 316.1 |
| 28 | 7 May 2021 | 10547 | 439.4 |
| 33 | 12 May 2021 | 12394 | 516.4 |
| 35 | 14 May 2021 | 12994 | 541.4 |
| 38 | 17 May 2021 | 14260 | 594.2 |
| 40 | 19 May 2021 | 15257 | 635.7 |
| 42 | 21 May 2021 | 16336 | 680.7 |
| 45 | 24 May 2021 | 18054 | 752.3 |
| 47 | 26 May 2021 | 19288 | 803.7 |
| 49 | 28 May 2021 | 20522 | 855.1 |
| 54 | 2 June 2021 | 22703 | 945.9 |
| 56 | 4 June 2021 | 23694 | 987.2 |
| 61 | 9 June 2021 | 26588 | 1107.8 |
| 66 | 14 June 2021 | 29519 | 1230 |
| 75 | 23 June 2021 | 34715 | 1446.4 |
| 89 | 7 July 2021 | 43101 | 1795.9 |
| 103 | 21 July 2021 | 51430 | 2142.9 |
| 117 | 4 August 2021 | 60309 | 2512.9 |
| 122 | 9 August 2021 | 63234 | 2634.8 |

Data show date of sampling, accumulated degree hours (ADH) calculated from hourly ambient air temperatures, and accumulated degree days (ADD) calculated from daily mean temperatures.

adjacent to the donors as well as from control plots located at least 2 m away and upslope from each donor. The selection of control sites 2 m away was to ensure they were not impacted by human decomposition products; this was based on previous research demonstrating limited ($< 1$ m) lateral transport of decomposition products through soil [6,10,24]. The five cores from each site were composited and stored at 4˚C. Soil samples were processed for pH, EC, and gravimetric moisture within 48 hours. A subsample of soils was frozen at -80˚C for elemental analysis.

## Soil physical analyses

In order to verify similarity of soil texture between experimental sites, particle size analysis (PSA) was performed on homogenized control soils from each site. Soils were air-dried and sieved using 2 mm sieves. Following the removal of organic matter, PSA was performed using a Malvern Mastersizer 3000 laser particle size diffractor. Clay mineralogy was performed on soils sourced from a section of the ARF not previously exposed to decomposition. Analysis of clay mineralogy was conducted according to methods outlined in Soukup et al. (2008) [38] and x-ray diffraction (XRD) analysis. Briefly, soil carbonates and exchangeable divalent cations were removed by repeated centrifuge washings of soil with a 1M pH 5 Na acetate buffer. Organic matter and Mn oxides were then removed by repeated additions of 30% $H_2O_2$ at 80˚C, followed by a four-hour digestion also at 80˚C. Fe oxides were then removed by the additions of 0.3M Na citrate and 1M Na bicarbonate, followed by the addition of Na dithionite at 75˚C. Soils were then Na-saturated, and particles $< 2\mu m$ (clay fraction) were isolated using Stoke's Law sedimentation. The clay fraction was saturated with K, Mg, and Mg-glycol and three slides were prepared for XRD: K-saturated at room temperature, Mg-saturated, and Mg-glycol-saturated. Additional K-saturated slides were prepared and heated to 300˚C and 550˚C. Diffractograms were created using a Brüker D8 goniometer with Ni-filtered, Cu Kα radiation, operating at 20kV and 5 mA. Scan ranges were 2–30˚2θ, with a step size of 0.05˚2θ, and step rate of 3 sec.

## Soil chemical analyses

Basic soil analyses followed methods detailed in Keenan et al. (2018a,b) [6,7]. Briefly, soils were hand-homogenized; rocks, insects, and large debris ($> 2$ mm) were removed. Gravimetric moisture was calculated from mass loss following oven drying for 72 h at 105˚C. Soil pH and EC were measured on soil slurries of 1:2 soil: deionized water using an Orion Star A329 multiparameter meter (ThermoScientific).

For elemental analysis we performed a water extraction designed to target dissolved organic and inorganic fractions present in the soil water solution; i.e., mobile and bioavailable elements. Water extractions also avoided some limitations of other extractants, including: 1) introducing pH changes during the extraction process that would be outside measured environmental parameters and potentially alter elemental solubility, 2) increasing yield by altering the solubilization and displacement of elements from the exchange and nonexchangeable soil fractions (Mehlich 3), and 3) determining total elemental concentrations irrespective of bio-availability (aqua regia digestion methods) [39]. In our extraction procedure 6.67 g soil was added to 40 ml of deionized water (1:7 soil:water) and shaken at 20˚C for four hours at 160 rpm. Following shaking, slurries were allowed to settle for one hour, and then were centrifuged at 2,175 rcf for two hours. Supernatants were filtered through 0.45 μm pore syringe filters, and final extractants were stabilized by acidification to 1% total volume (pH $< 2$) with spectrophotometric grade nitric acid. Elemental analysis of soil extracts was done by inductively-coupled argon plasma optical emission spectroscopy (ICPOES). Analyses were performed at the

University of Tennessee's Water Quality Core Facility laboratory on a ThermoElement ICAP7400 fitted with polyethylene vials and tubing and a Teflon nebulizer to minimize background Na and B cross-contamination from borosilicate glass components. This method was based on EPA method 200.7 [40]. Analytical batches included nitric acid blanks and quality control reference standards within the concentration range of the study samples. Reference standard concentrations were within 90–100% of nominal values.

Element selection for analyses was based upon human body composition in combination with extraction suitability for ICPOES analysis. Elements selected for analysis included: Ca, P, K, S, Na, Mg, Fe, Cu, Mn, Zn, Se, Co, and B. The mobility of Al (as well as Fe, Cu, Mn, Zn, and Co) can alter under soil acidification frequently observed during human decomposition, thus Al was included as an element of potential interest, both in terms of utility for human remains detection and for evaluating its solubility patterns in decomposition environments.

Raw data is available as a supplemental dataset (S1 File).

### Statistical analyses

Human donors (n = 3) were treated as experimental replicates. Kruskal-Wallace nonparametric tests were performed to test for differences between decomposition impacted and control soils over the course of the study ($p < 0.05$), followed by Welch T-tests conducted at individual sampling time points. Pearson correlation coefficients were used to construct a correlation matrix comparing abiotic and elemental parameters in decomposition soils. Principal component analysis was performed to visualize multivariate relationships between abiotic parameters and elements throughout the study. For multivariate cluster analysis, elemental concentrations were scaled to a mean of zero and standard deviation of 1 and hierarchically clustered by Pearson correlations. All analyses and visualizations were performed in R (version 3.6.1) using the tidyverse (1.2.1), vegan, ggplot2 (version 3.2.1), pheatmap (1.0.12), and RColorBrewer (version 1.1–2) packages [41–46]. R code and associated analyses files are available: https://github.com/jdebruyn/SPOT-soilelements.

## Results

### Decomposition progression

Cool temperatures at the beginning of the study resulted in a prolonged period of early decomposition in which little change was evident. Bloating and initial decomposition fluid leaching was apparent by day 14, as both temperatures and insect activity increased. Active decay began by day 17, followed by the expansion of "wet" cadaver decomposition islands (CDIs) throughout days 28 to 42 and during the period of greatest tissue loss. CDIs were considered dry by day 80 and contained hard crusts with visible adipocere. Donors exhibited either late advanced decay/very early skeletonization (moderate tissue remaining) or mummification by the end of the study (day 122).

### Soil texture

Soils for all donor sites were classified as silt loam and composed of the following fractions: sand (26 to 41%), silt (56 to 69%), and clay (3 to 5%).

### Soil mineralogy

X-ray diffractograms reflect a similarity in clay mineral composition between soil sites (S1 Fig). A characteristic shift in d-value from 1 nm under $K^+$ saturation to 1.4 nm under both $Mg^{2+}$ and $Mg^{2+}$ + glycol saturation indicated the presence of vermiculite [34,47]; the presence

of a 1.23 nm peak under $Mg^{2+}$ and $Mg^{2+}$ + glycol saturation indicated the presence of vermiculite interstratification with mica. Kaolinite (d = 0.7 nm) was also present.

## Soil biogeochemistry

The pH and EC of soils impacted by human decomposition significantly differed from controls over the entire study (Kruskal-Wallace, p < 0.05) (Table 3). As decomposition products moved into the soil, pH decreased and EC increased, both of which became significantly different relative to controls on day 33 (Fig 1, S1 and S2 Tables). Conductivity reached a maximum on day 117 at 455.9 ± 201 µS cm$^{-1}$. Soil pH reached a minimum of 5.8 on days 45 and 49, and remained significantly different than controls at the end of the study.

With the exception of B, all elemental concentrations differed significantly between decomposition and control soils (Kruskal-Wallace, p < 0.05) (**Table 3**). Principal component analysis revealed three groupings of elements: Group 1 elements increased early and persisted at elevated concentrations through the end of the study; group 2 elements increased early but decreased at later time points, and group 3 elements increased only at later time periods (Fig 2). The grouping of elements is also supported by hierarchical cluster analysis of elemental profiles (Fig 3). Hierarchical clustering further revealed that while groups 1 and 3 formed distinct clusters, group 2 elements, particularly those of Mn and Se, were more closely related to group 1 elements.

Group 1 contained the elements Na, S, P, and K. The elements P, K, and S reached mean peak concentrations and significantly differed from control soils between days 33 and 56 with soil concentration increases that ranged from 3.7 to 22 times those of control concentrations. Na exhibited the greatest increase above controls, followed by P, then S, then K. Na concentrations reached a first maximum on day 38 at 213.2 ± 82.3 µg gdw$^{-1}$, and a second maximum concentration on day 117 at 232.7 ± 161.6 µg gdw$^{-1}$, approximately two orders of magnitude

**Table 3. Effects of decomposition on element abundances in soil.**

| Element | Control | Decomposition | $X^2$ | p |
|---|---|---|---|---|
| pH | 7.4 ± 0.2 | 6.4 ± 0.5 | 81.893 | <**0.001**\*\*\* |
| Conductivity (EC) | 70.8 ± 17.5 | 266.7 ± 118.4 | 59.281 | <**0.001**\*\*\* |
| Calcium (Ca) | 137.9 ± 17.2 | 210.1 ± 80.8 | 13.149 | <**0.001**\*\*\* |
| Phosphorus (P) | 1 ± 0.1 | 9.8 ± 7.4 | 50.645 | <**0.001**\*\*\* |
| Potassium (K) | 27.6 ± 4.8 | 80 ± 35.5 | 50.597 | <**0.001**\*\*\* |
| Sulfur180 (S) | 6.5 ± 0.8 | 23.4 ± 10 | 76.701 | <**0.001**\*\*\* |
| Sodium (Na) | 2.3 ± 0.4 | 126.7 ± 70.5 | 113.15 | <**0.001**\*\*\* |
| Magnesium (Mg) | 7.5 ± 0.7 | 19.2 ± 7.7 | 61.467 | <**0.001**\*\*\* |
| Iron (Fe) | 2.2 ± 0.7 | 6.2 ± 7.3 | 14.163 | <**0.001**\*\*\* |
| Copper (Cu) | 0.05 ± 0.01 | 0.08 ± 0.04 | 16.306 | <**0.001**\*\*\* |
| Manganese (Mn) | 0.6 ± 0.2 | 19 ± 15.9 | 62.469 | <**0.001**\*\*\* |
| Zinc (Zn) | 0.04 ± 0.01 | 0.15 ± 0.12 | 60.577 | <**0.001**\*\*\* |
| Selenium (Se) | 0.01 ± 0.00 | 0.03 ± 0.02 | 60.213 | <**0.001**\*\*\* |
| Cobalt (Co) | 0.00 ± 0.00 | 0.08 ± 0.11 | 52.765 | <**0.001**\*\*\* |
| Boron (B) | 0.36 ± 0.04 | 0.53 ± 0.18 | 3.351 | 0.067 |
| Aluminum (Al) | 9.1 ± 2.3 | 11.6 ± 13.8 | 5.196 | **0.023**\* |

Control and decomposition soil means ± standard deviations (SD) (n = 3) across all timepoints analyzed are presented, along with results ($X^2$ and p value) of Kruskal-Wallace tests comparing decomposition soils with controls over the entire trial (df = 1). Units of conductivity (EC) are µS cm$^{-1}$, and units for all soil elemental concentrations are presented in µg gdw$^{-1}$ soil. Elements (excepting Al) are listed in approximate order of percentage found in the human body from greatest to least.

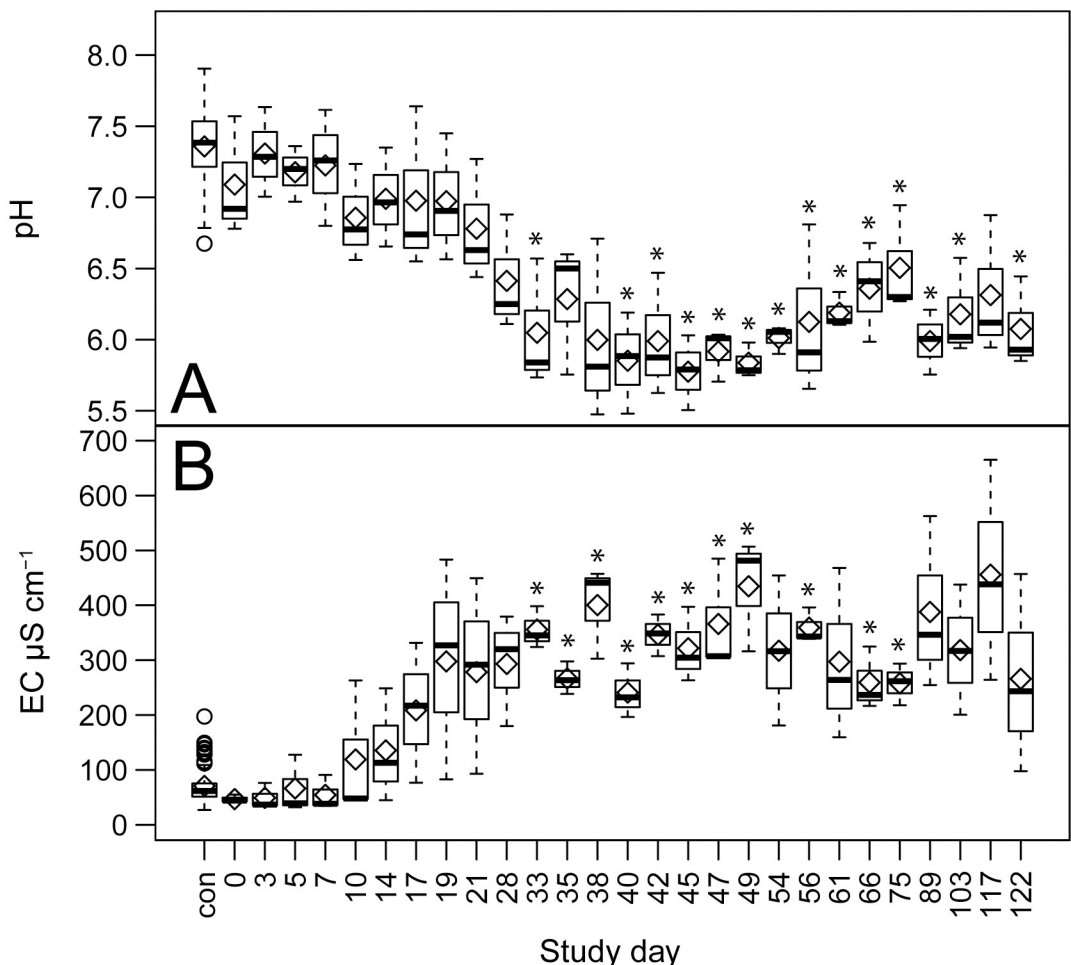

**Fig 1. Effects of decomposition on soil pH and conductivity.** Comparisons between decomposition impacted and control soils over the entire study trial for A) pH, and B) electrical conductivity (EC). Data are means (diamonds) and medians (bars) for n = 3 replicate donors. Statistically significant ($p < 0.05$) differences between decomposition and control soils are denoted by an asterisk.

greater than control soil concentrations, and significantly differed from control soil intermittently between days 35 through 49 (Fig 4, S1 and S2 Tables). Overall, group 1 elements were highly positively correlated with each other (Pearson r = 0.73 to 0.89), with Na and K exhibiting the highest correlation (Pearson r = 0.89) (Fig 5). This group of elements was also positively correlated with EC (Pearson r = 0.59 to 0.85) with the highest correlation occurring between EC and Na, and the lowest between EC and P. All group 1 elements were negatively correlated with pH (Pearson r = -0.38 to -0.5). Soil charge concentrations of exchangeable cations ($Na^+$ and $K^+$) also exceeded those found in control soils. The mean maxima for $Na^+$ (days 38 and 117) were 9.3 ± 3.6 $\mu mol_c$ $gdw^{-1}$ and 10.7 ± 7 $\mu mol_c$ $gdw^{-1}$ respectively, an increase of approximately 90 times, whereas maxima for $K^+$ occurred between days 33 through 56 at a mean of 3.1 $\mu mol_c$ $gdw^{-1}$ an increase of about 4 times that of control soils (Fig 6, S3 Table).

Group 2 contains the elements Mn, Mg, Se, B, and Ca, of which Mn, Se, and B are considered trace elements in the human body. Of this group, elemental concentrations increased early, and with the exception of B, maximum mean elemental concentrations were recorded on day 49 (Figs 3 and 7, S1, S2 and S4 Tables). Mg and Ca concentrations were highly

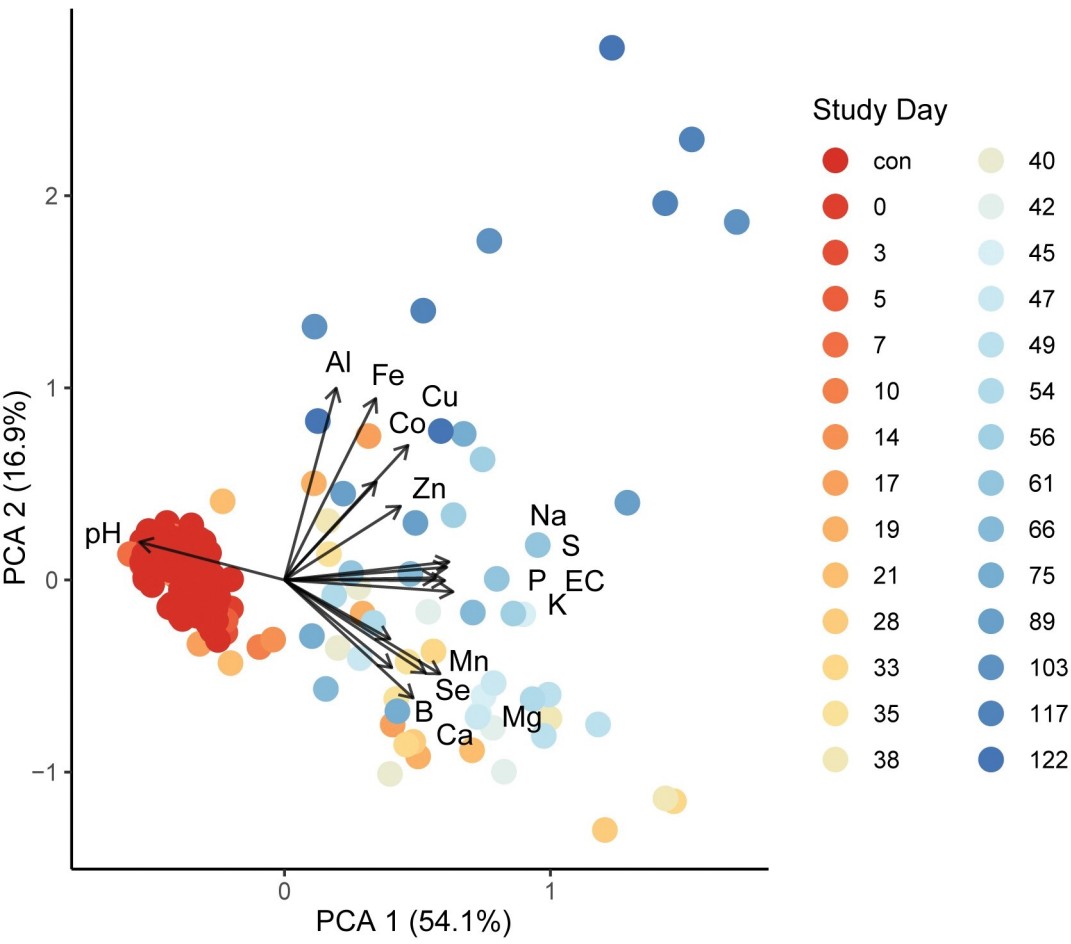

**Fig 2. Principal component analysis (PCA) of pH, conductivity (EC), and elemental concentrations by study day.** Points represent individual samples colored by study day, and vectors represent elements, pH, and EC. The PCA diagram shows three groups of elements. Group 1: Sodium (Na), sulfur (S), phosphorus (P), EC, and potassium (K). Group 2: Manganese (Mn), selenium (Se), boron (B), magnesium (Mg), and calcium (Ca). Group 3: Aluminum (Al), iron (Fe), cobalt (Co), copper (Cu) and zinc (Zn).

correlated (Pearson r = 0.89; Figs 3 and 5), and after reaching maximum concentrations, decreased on day 54 by approximately 40%, after which concentrations remained relatively stable and intermittently elevated above background levels through day 75. Mn and Se soil concentrations did not significantly differ from controls on the study date at which their maximum mean concentrations were observed (day 49); however, in the case of Se, significant elevations were observed between days 61 through 89 and again on day 117, Mn differed from background soils on day 89, and reached maximum concentrations of $62.3 \pm 43.2$ µg gdw$^{-1}$, 89 times that of controls. Mg, Se, B, and Ca had maximum concentrations of 3.1 to 17.8 times those found in control soils. All elements in this group were positively correlated with EC (Pearson r = 0.24 to 0.59), although these correlations were weaker than observed for group 1 elements (Fig 5). Similar to group 1, group 2 elements were negatively correlated with pH (Pearson r = -0.32 to -0.67). Soil charge concentrations of exchangeable cations (Ca$^{2+}$ and Mg$^{2+}$) also exceeded those found in control soils, and the mean maxima for both elements (day 49) were $20 \pm 6.3$ µmol$_c$ gdw$^{-1}$ and $3 \pm 0.1$ µmol$_c$ gdw$^{-1}$ respectively, an increase of between 3 and 5 times that of control soils (Fig 6, S3 Table).

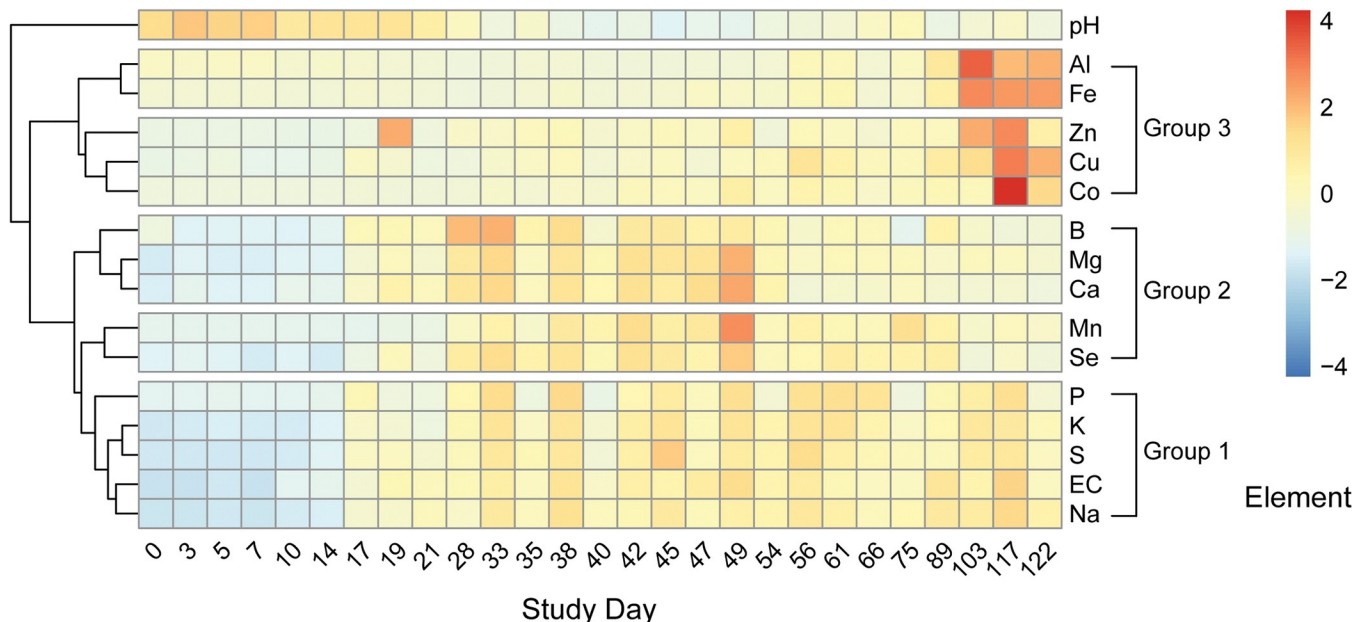

**Fig 3. Heatmap showing scaled changes in elemental concentrations by study day.** Three distinct groups and patterns of change occur in EC and elemental concentrations. Group 1 includes sodium (Na), conductivity (EC), sulfur (S), potassium (K), and phosphorus (P); these increase by day 17 and soil concentrations persist throughout the study. Group 2 includes selenium (Se), manganese (Mn), magnesium (Mg), calcium (Ca), and boron (B); these elements also increase by day 17, however soil concentrations decline later in the trial. Group 3 includes: cobalt (Co), copper (Cu), zinc (Zn), aluminum (Al), and iron (Fe); these elements increase following day 89. Elemental concentrations are scaled to a mean of zero and standard deviation of 1 and are hierarchically clustered according to Pearson correlations.

Group 3 consisted of the elements Fe, Co, Cu, and Zn, all considered trace elements within the human body. Increased concentrations in decomposition soils were primarily observed later in the study, with most elements exhibiting significant differences from controls on or following day 45, and maximum mean soil concentrations between days 103 and 117 (Figs 3 and 8, S2 and S4 Tables). The most significant increases with respect to controls occurred in Fe at $27.3 \pm 4.2$ μg gdw$^{-1}$. Al, not an element found in the human body but present as Al oxy-hydroxides in soils [34], exhibited significantly decreased concentrations in comparison with controls ($> 50\%$ decrease) prior to day 75, and increases of an order of magnitude at maximum concentrations on day 103 at $58.9 \pm 15.8$ μg gdw$^{-1}$. Of this group of elements only Al and Fe exhibited a high positive correlation (Pearson $r = 0.82$) (Figs 3 and 5). Correlations between group 3 elements and EC showed a similar range as those for group 2 elements (Pearson $r = 0.20$ to $0.60$), however correlations between group 3 elements and pH were weaker in comparison with groups 1 and 2 (Pearson $r = 0.01$ to $-0.38$).

## Discussion

Our study evaluated decomposition-induced changes in a suite of elements in soil solutions that included both those occurring in significant quantity in the human body (Ca, P, K, S, Na, Mg) as well as biologically important trace elements (Fe, Cu, Mn, Zn, Se, Co, B). Al was included with this study because it is commonly found in soils and its mobility in the soil solution might be affected by soil acidification, which is frequently observed during human and less often during animal decomposition. We revealed significant increases in all elements except for B and demonstrated that dissolved soil elements could be clearly ordered into three groups according to their temporal patterns during decomposition.

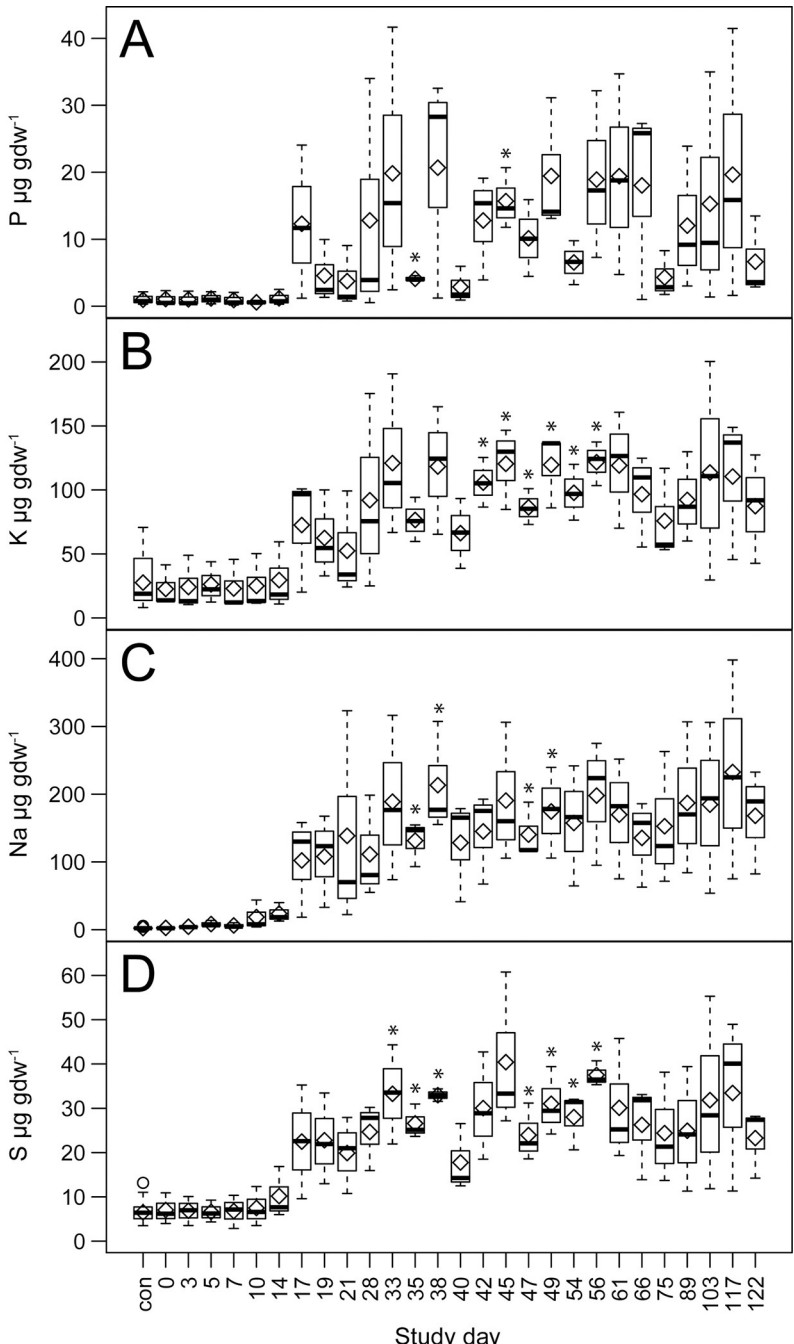

**Fig 4. Effects of decomposition on group 1 element abundances in soil.** Comparisons between decomposition and control soils over the entire trial for A) phosphorus, B) potassium, C) sodium, and D) sulfur. Data are means (diamonds) and medians (bars) for n = 3 replicate donors. Statistically significant ($p < 0.05$) differences between decomposition and control soils are denoted by asterisks.

## Group 1 elements

Group 1 elements consisted of those found in high abundance in the human body (P, K, Na, and S) [29]. Concentrations of these elements in soil all increased initially by the onset of active decay and were closely coupled with increases in EC, and to a lesser extent decreases in pH.

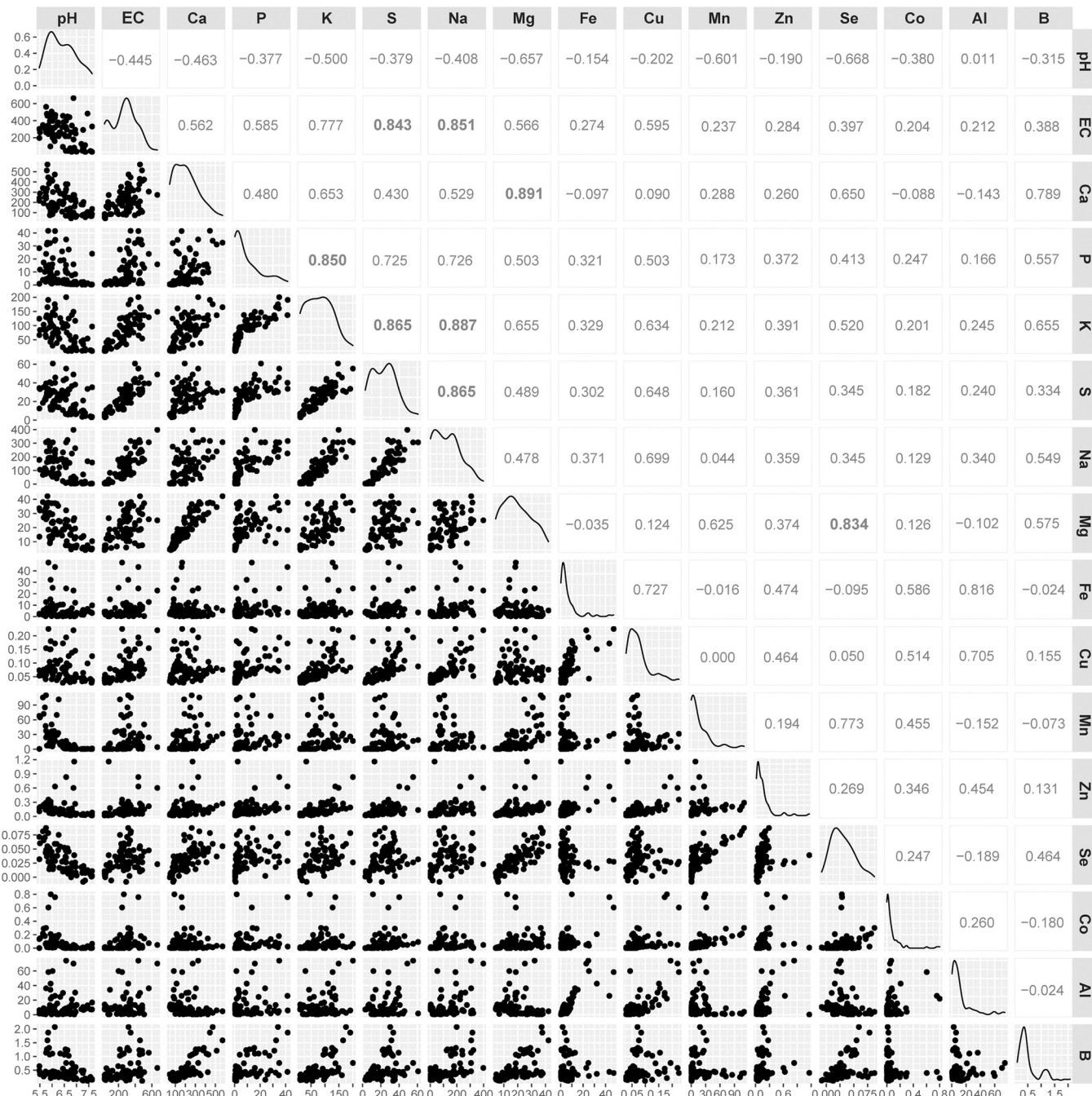

**Fig 5. Correlations between soil pH, EC, and elemental concentrations during human decomposition.** Pearson correlation matrix showing relationships between pH, conductivity (EC), and individual elements in decomposition-impacted soils. Diagonal panels display pH, EC, and individual elemental distributions. Pearson correlation r values are shown between pH, EC, and elements at the top of the matrix. Pearson correlation r > 0.80 are shown in large bold type. EC and elemental concentration ranges are presented in $\mu S\ cm^{-1}$ (EC) and $\mu g\ gdw^{-1}$ soil (elements).

The presence of P in decomposition environments has arguably received the greatest characterization after C and N. P is the second-most abundant of the non-trace mineral elements tested that originate from the living adult human body (estimated 0.7 to 1.2% of weight) after Ca [29,48,49]. Approximately 85% of P in the body is found in the skeleton in the form of

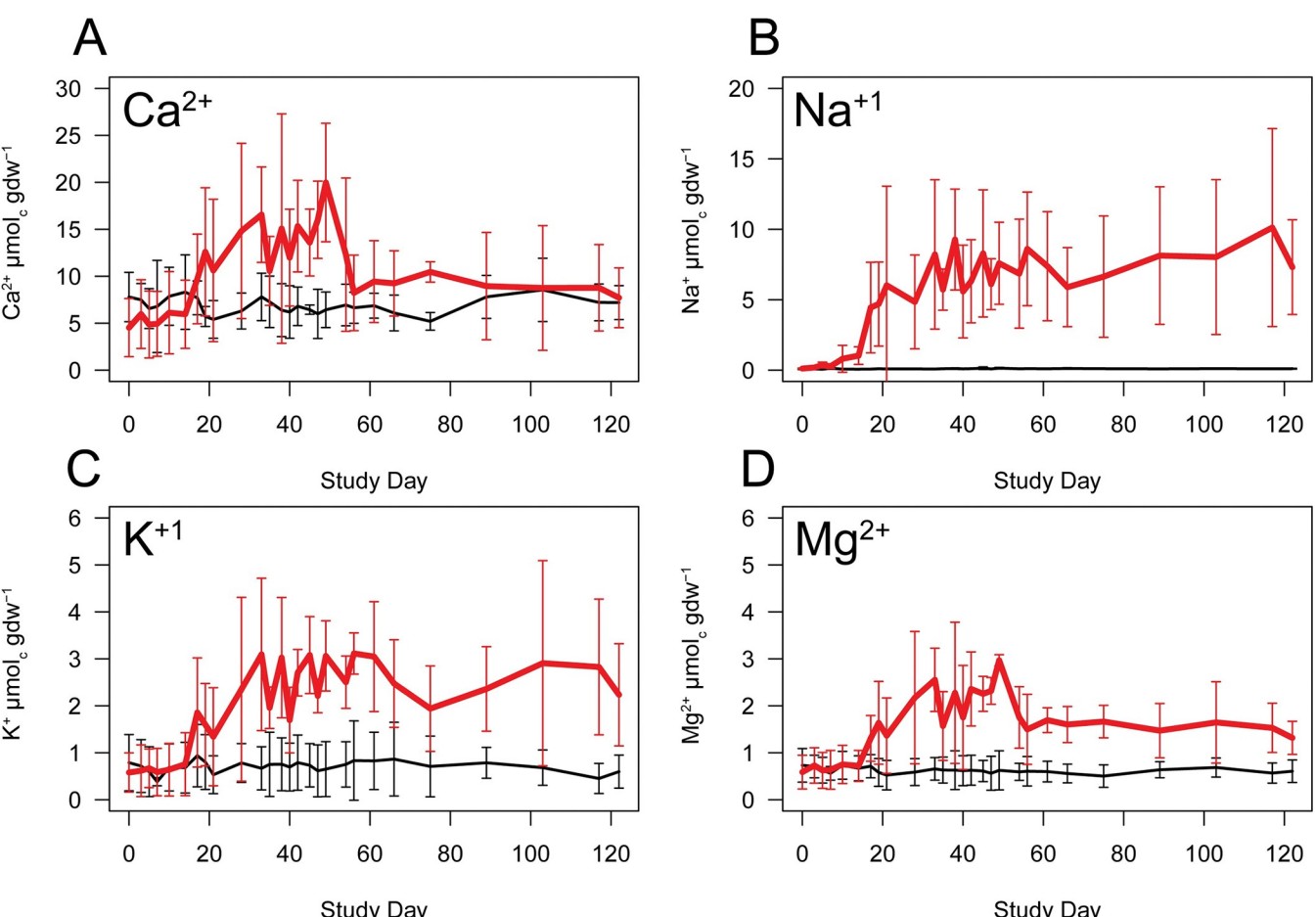

**Fig 6. Effects of decomposition on charge concentrations in soil.** Comparisons between molar charge concentrations in decomposition and control soils over the entire trial for A) calcium, B) sodium, C) potassium, and D) magnesium. Selected elements are exchangeable base cations in soil. Data are means ± standard deviations for n = 3 replicate donors.

hydroxyapatite, and the remaining 15% is found in soft tissues, viscera, and the extracellular space [48,49]. For our mean cadaver mass of 78.9 kg, this constitutes between 552 and 947 g of total P in the body, with approximately 83 to 142 g in the soft tissues that could be released during early decomposition. Previous studies have reported concentrations of P that are variously described according to fractions selected for quantification (total P, lipid P, bioavailable P, etc.) which differ according to their extraction methodology, therefore direct comparisons of P between studies is problematic. For the purposes of this discussion "soil P" does not distinguish between fractions unless specified, nor does it imply that detectable quantities are dissolved. Animal studies consistently report initial increases in soil P across multiple extraction fractions at the onset of decay [15,16,18,19] as early as day 8 [18] and 190 ADD [16]. Following the initial increase, P generally decreased gradually but remained elevated for months or years under animals > 1.5 kg [12,13,18,21,22,24,25,28]. Due to difficulties associated with replication in human decomposition studies, soil P has not been characterized to the same extent for human decomposition, especially longitudinally, so patterns are unclear. For example: while we observed an initial increase in P during early decomposition, accompanied by decreased pH, concentrations of P exhibited a high degree of variance and did not statistically differ from background values except on two dates (days 35 and 45) with maximum values occurring on

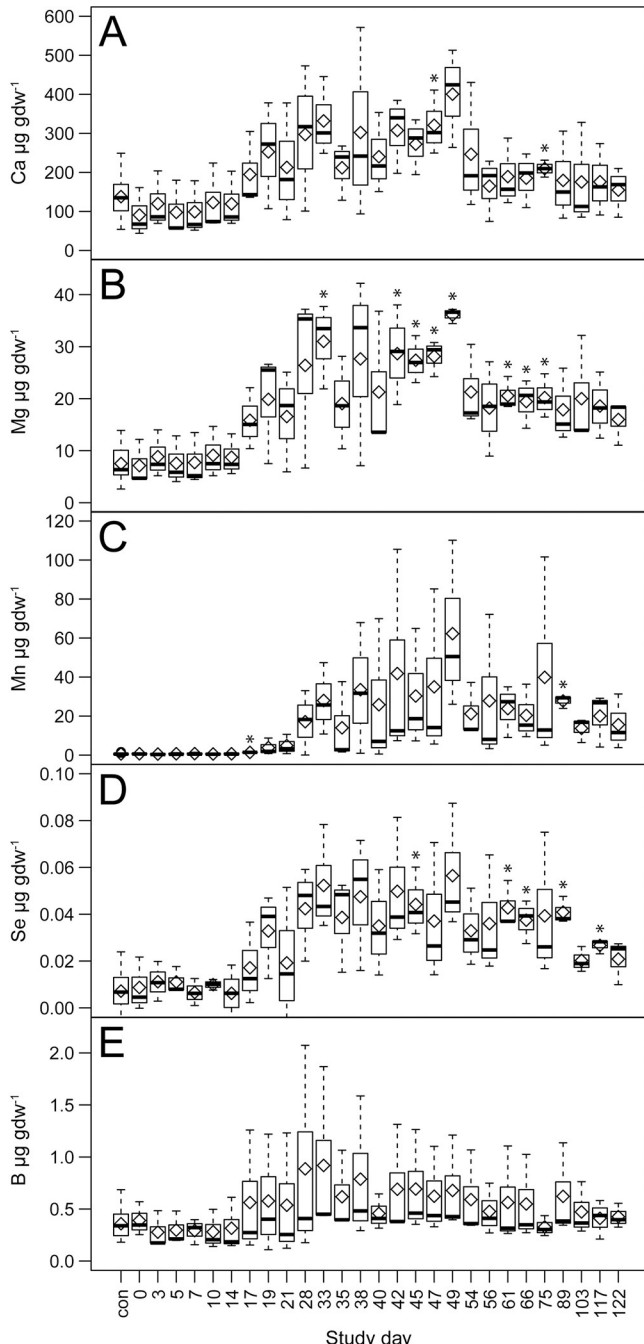

**Fig 7. Effects of decomposition on group 2 element abundances in soil.** Comparisons between decomposition and control soils over the entire trial for A) calcium, B) magnesium, C) manganese, D) selenium, and E) boron. Data are means (diamonds) and medians (bars) for n = 3 replicate donors. Statistically significant ($p < 0.05$) differences between decomposition and control soils are denoted by asterisks.

day 38; these results differ slightly from those of Cobaugh et al. [5] who reported an overall lack of significant changes in both P and pH at the same facility. In a cross-sectional study, Fancher et al. [17] found elevated P up to 1752 days postmortem over two soil series, also noting that pH did not display specific trends. Likewise, Szelecz et al. [26] reported elevated P

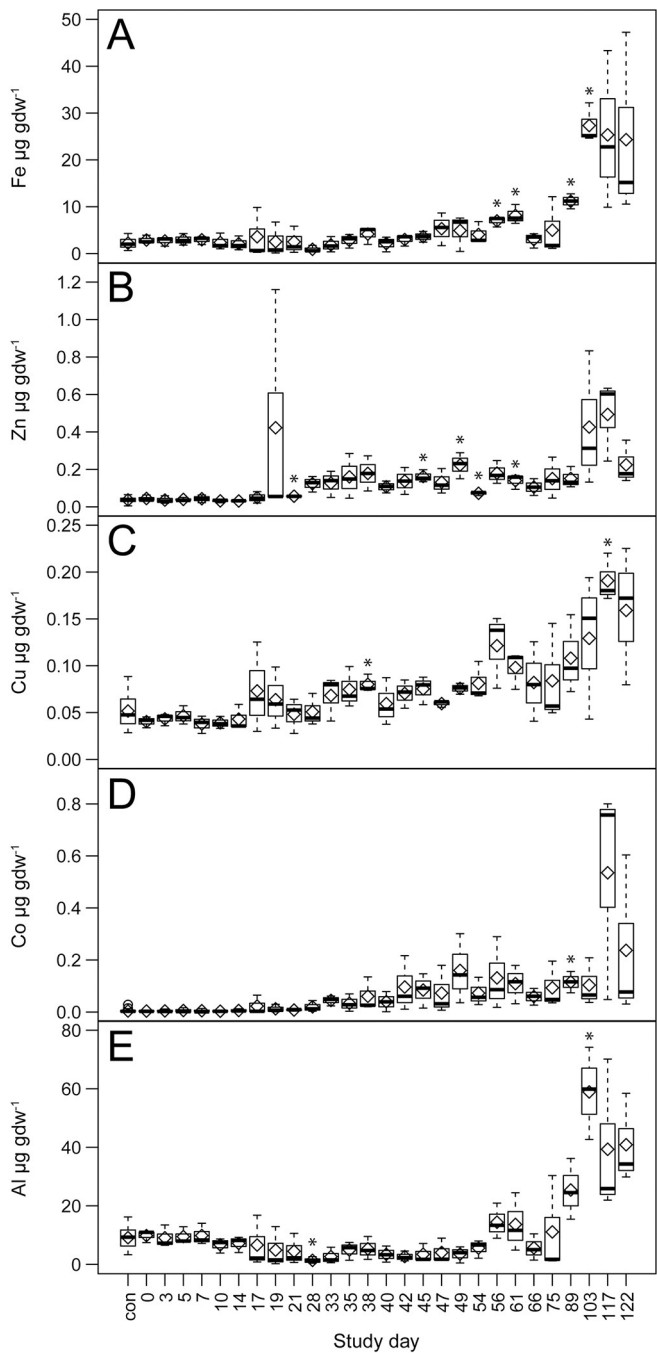

**Fig 8. Effects of decomposition on group 3 element abundances in soil.** Comparisons between decomposition and control soils over the entire trial for A) iron, B) zinc, C) copper, D) cobalt, and E) aluminum. Data are means (diamonds) and medians (bars) for n = 3 replicate donors. Statistically significant ($p < 0.05$) differences between decomposition and control soils are denoted by asterisks.

with no change to soil pH in a forensic case estimated at 22 months in which dry remains were recovered. Aitkenhead-Peterson et al. [10] used a water extraction and reported elevated P after 248 days (>5400 ADD) in soils that acidified under two human donors. Inorganic P exhibits complex soil chemistry (as predominantly $H_2PO_4^-$ in our soil pH range), strongly adsorbing to Al and Fe oxyhydroxides and to clay minerals, and is thus retained by soil; the

solubility of both Al and Fe in the soil solution are pH-dependent, notably increasing at pH < 6 [34]. Taken together, this suggests some degree of complexation of unbound forms of inorganic P. Since P is commonly found in decomposition soils, it is likely that early measurements may reflect dissolved organic forms of P from the degradation of serum, tissues, and lipids, which have not sufficiently decomposed to complex with soil. Late measurements (skeletonization and dry remains) may be more reflective of inorganic P sourced from bone, and given the greater quantities of P available in the bone matrix it is likely that elevated soil P concentrations derived from bone would be found outside the time frame of our study.

Human bodies are composed of approximately 0.15–0.18% K and 0.11–0.14% Na by weight [29,48]. Both elements are generally present as uncomplexed cations (as $Na^+$ and $K^+$) and largely found in extra- and intracellular fluid, functioning jointly to regulate blood pressure, fluid, and electrolyte balances [48]. For our mean cadaver mass of 78.9 kg, this constitutes up to 142 g of $K^+$ and 110 g $Na^+$ potentially entering the soil as free cations with the early flush of decomposition fluid. As decomposition progressed, we observed soil concentrations of both $K^+$ and $Na^+$ that exhibited similar temporal patterns of change, increasing with EC during active decay and remaining significantly elevated into early advanced decay. However, despite the similar concentrations in the human body, mean soil and molar charge concentrations of $K^+$ were approximately half those of $Na^+$. $Na^+$ and $K^+$ are exchangeable cations in the soil, and $K^+$ is selectively preferred to $Na^+$ on the soil exchange complex [34]. It is possible that this overall reduction in $K^+$ in comparison with $Na^+$ may result from an exchange complex partially saturated by both $K^+$ and the large amounts of exchangeable ammonium ($NH_4^+$) released during early decay [5,7–9,11,14,15,18]. It is interesting to note that our soils are partly composed of vermiculite and interstratified mica-vermiculite, the interlayer of which can collapse when saturated with $K^+$ and $NH_4^+$. If this phenomenon occurs, it would then immobilize any $K^+$ within the mineral interlayers and effectively reduce or eliminate any further exchange processes.

Overall, our time-resolved results for K and Na concentrations are consistent with reports from other human decomposition studies [9,10,17]. For example, in a longitudinal study at the ARF, Vass et al. [9] assessed soil elemental changes for a variety of soil cations (including Na and K), and reported that collectively elemental increases tended to peak at about 750 ADD, followed by a second peak at 3750–4000 ADD. In our study, both elements reached their first maxima between days 38–56, corresponding to the period 594 to 987 ADD, however we did not observe appreciable decreases in concentrations. Other studies of decomposing human subjects have also reported broadly elevated Na and K in soil that range from months to years [10,17]. Animal decomposition studies have shown no consensus on soil K patterns [16,18,22,25,28]. However, animal studies have shown evidence of generally elevated soil Na during decomposition [25] as well as elevated Na in the tissues of nearby plants [16]. While our study site was a forest with ample nearby vegetation, we did not observe an appreciable decrease in soil Na over the course of our study, suggesting that vegetation uptake (if any) did not necessarily constitute a major loss pathway for Na. Both Na and K conjugate with fatty acids derived from fat decomposition, forming fatty acid salts and contributing to the formation of adipocere, potentially immobilizing a portion of both elemental pools. The eventual breakdown of these fatty acid salts may constitute a slow release of both elements contributing to later elevated concentrations, however in one study examining $Na^+$ and $K^+$ fatty acid salts, neither were found to change over a period of several months [30], suggesting that this might be an impact observed much later in the decomposition time frame.

Sulfur quantities in the human body are roughly equal to that of K (0.15%) [29]; S is a constituent of the amino acids methionine and cysteine, sulfur-containing acids (taurine and homocysteine), and bile salts resulting from the conjugation of bile acids with taurine.

Interestingly, while temporal patterns in soil were the same as that of K in our study, soil concentrations of S were lower by approximately two-thirds. This is likely because the mobility and loss pathways for S differ considerably from the base cations Na and K, and for P. Unlike the other group 1 elements, S is primarily tied up in organic molecules that require microbial degradation to more available forms. S-containing VOCs have been detected during decomposition [1,50–53], thus a fraction of S is lost to the atmosphere. Elevated soil S concentrations were reported in both animal [25,54] and human decomposition studies [9,10]. Recent work has also shown the presence of S-containing metabolites, notably taurine and other amino acids, in decomposition-impacted soils [8]. Since this mobility in soil originates primarily from microbial processes and will differ from elements that interact with the soil cation exchange complex, it is possible that the systematic degradation of S-containing compounds may show promise as temporal indicators of decomposition progression.

Taken together, our results suggest that initial increases in the group 1 elements Na, K, P, and S stem directly from cadaver-derived soil inputs, and that both their concentrations and persistence in soil varies based upon soil complexation (P), the dynamics of the soil exchange complex (Na, K) and gradual releases attributable to microbial degradation (S).

## Group 2 elements

Group 2 contains the elements Ca, Mg, Mn, Se, and B, of which Mn, Se, and B are considered trace elements in the human body [48,49]. Soil concentrations for elements in this group increased at the same time as group 1 elements, but then declined in late active and early advanced decay.

Ca comprises approximately 1.5–2% of the human body weight. More than 99% of the Ca in the body is found in the skeleton in the form of hydroxyapatite [29,48,49]. In contrast, Mg estimates vary between 25 g and 0.02% of the human body, and is found distributed between skeletal and soft tissues in approximately equal quantities [29,48]. In a 78.9 kg human the total amount of Ca and Mg would be about 1.2 to 1.6 kg (12 to 16 g in soft tissue) and 16 to 25 g (8 to 12.5 g in soft tissue), respectively. We observed highly correlated patterns of increase in Ca and Mg during the period of soft tissue loss (active and early advanced decay), and sudden decreased soil concentrations of both elements as tissue loss slowed during the transition from active to advanced decay. Surprisingly, soil concentrations of Ca and Mg were higher than would be expected based on input quantities from soft tissue. $Ca^{2+}$ and $Mg^{2+}$ are exchangeable cations in soil and dominate the exchange complex of circumneutral pH soils. In our study, combined monovalent cation charge in control soils ($Na^+$ and $K^+$) was approximately 10% of combined divalent cation charge ($Ca^{2+}$ and $Mg^{2+}$). In decomposition-impacted soils the percentage of combined monovalent charge rose to 57% on day 17, and by day 56 exceeded 100% of divalent cation charge. During early decomposition, high concentrations of exchangeable $NH_4^+$ resulting from soft-tissue degradation are released into the soil in conjunction with $K^+$ and $Na^+$; it is possible that the combined charge of these three monovalent cations could shift the exchange equilibrium prior to day 56, releasing previously adsorbed $Ca^{2+}$ and $Mg^{2+}$. This potential release of divalent cations from the exchange complex may explain both the persistence of elevated $Na^+$ and $K^+$, as well as the presence of early increased concentrations of Ca and Mg that could not otherwise be attributed to soft-tissue release or to much later-occurring bone diagenesis. This release would also be expected given the strong preference of vermiculite for $NH_4^+$ and $K^+$ [34]. As observed for K and Na, Ca and Mg exhibit similar temporal patterns of change in human decomposition-impacted soils, however reports vary from non-specifically elevated [10,17] to observing increases only at late time points following advanced decay [9,30]. Ca is the most prominent (greatest abundance) of the two elements and it has been

proposed as a good candidate for inclusion in soil-based decomposition models as a predictor of PMI [17]. Animal studies have reported similar increases in Ca and Mg [16,25]. Our results did not show increases of either element late in decomposition that could be potentially attributed to bone material; given our study length of 122 days, we did not expect appreciable bone degradation. Our results suggest that while both Ca and Mg are plentiful in the human body, their immediate accessibility during active decay (soft tissue loss) is unpredictable due to differences in a given soil's cation exchange dynamics.

Interestingly, soil solution concentrations of both Ca and Mg were reduced beginning on day 56 during the period in which active decay was transitioning to advanced decay. At this time the CDI was still "wet" and exhibiting evidence of fat decomposition products. Shortly thereafter (day 80) CDIs were observed to have dried and formed hard crusts in which visible adipocere was present. The formation of adipocere includes interactions between fatty acids, Na, and K, leading to the formation of fatty acid salts, and there is evidence that Ca and Mg present in the soil can displace Na and K, thus forming Ca and Mg salts [55,56]. Given that our extraction methodology was water-based and thus would not solubilize adipocere, it is entirely possible that we have observed dual phenomena: the evolution of Ca and Mg from the soil exchange complex and into the soil solution, followed by the incorporation of both cations into adipocere. This possibility is supported by Lühe et al. [30], in which both Ca and Mg fatty acid salt concentrations increased over time following human decomposition. Further study would be needed to quantify fatty acid and fatty acid salt content of these soils to directly link elemental changes with fat decomposition transformation products, and to begin the creation of an elemental soil budget.

Mn is considered a trace element in humans, and is distributed throughout a variety of tissues and in serum with estimated abundances of less than 20 mg [48,49]. Increases in Mn in decomposition soils were notable: an overall increase of 89 times that found in control soils, and the largest increase in comparison with controls within the group 2 elements. Soil acidification is known to increase solubility of Al, Mn and Fe [57]. Soils in our study gradually acidified through day 45 to a mean pH of 5.8, suggesting that the substantial concentration increases in Mn found in decomposition soil may derive in part from cadaveric inputs, but also from the soil matrix due to an increase in pH-induced solubility. Soil acidification in combination with reduced oxygen found in decomposition environments are likely to favor $Mn^{2+}$ from solubilized Mn [7,34].

Se is also considered a trace element within the human body, generally found as a component of selenoproteins, and with serum or plasma concentrations ranging from 61–99 $\mu g\,L^{-1}$ [58]. While soil increases of Se in this study were not substantial in comparison with other elements, it is worth noting that Se behaves similarly in many respects to S, both in the human body and in the environment [34,48,58]. Additionally, Se found within plants is directly related to the Se content within soil [58]. Thus, it is possible that fate and behavior of Se in soil and its subsequent uptake by nearby plants may prove a uniquely useful indicator of decomposition progression or presence.

In contrast with elemental grouping levels derived from our PCA ordination, hierarchical clustering based upon correlative relationships suggested an alternative view that Mg, Ca, Mn, Se, and B are related to group 1 elements rather than forming their own unique grouping level. Correlative relationships are mathematically strong with respect to elemental concentration changes and their timings, however caution should be exercised in interpretation to avoid conflating apparent patterns derived from a combination of tissue decomposition, pH, and the soil exchange. For example, unlike $Ca^{2+}$ and $Mg^{2+}$ which are highly correlated and behave similarly in soil, Mn and Se appear to derive from differing sources (soil and soft tissue, respectively) and by dissimilar mechanisms, thus suggesting that their correlation was largely

mathematical rather than mechanistic. It is our view that these elements (Mg, Ca, Mn, Se, and B) can be classified, albeit loosely, into their own separate grouping level.

Taken together, increases in the group 2 elements appear to result partially from cadaver-derived materials (Se) but also potentially from the impacts of the influx of group 1 elements (Na, K and ammonium) on the soil exchange complex (Ca, Mg), or from increased solubility resulting from soil acidification (Mn).

### Group 3 elements

Group 3 consists of Fe, trace elements in the human body (Cu, Zn, Co), and Al. Unlike groups 1 and 2, this group of elements exhibited increased concentrations at the end of the decomposition period after the majority of tissue mass had been lost. Cu and Zn also exhibited slight increases between days 19 through 66. Approximately 4 g of Fe is present in the human body, about two-thirds of which is found in hemoglobin, and 15% associated with muscle tissue and enzymes [49]. It is estimated that about 100 mg of Cu is present in the human body, primarily in skeletal and muscle tissues, and enzyme cofactors [49]. Zinc in the human body ranges from approximately 1.5 to 2.5 g, most of which is located in skeletal and muscle tissues [48]. Cobalt constitutes around 1 to 2 mg in the human body, and is a component of vitamin $B^{12}$, coenzymes, and otherwise distributed throughout a variety of tissues [59]. Al is toxic to humans although transiently found in the body in small quantities. Al has been included in this elemental survey because it is present in soil in numerous aluminosilicate minerals as well as Al oxyhydroxides, and can become mobile in soil under soil acidification that is frequently associated with human decomposition. Given their trace abundances in the human body it is likely that as the soil slowly acidified and remained acidic, these elements were gradually solubilized from soil minerals, and that late soil increases in these elements results less from cadaver tissue-derived inputs and more from the soil itself.

### Conclusions

This work presents a detailed longitudinal characterization of changes in soil solution elemental concentrations during human decomposition furthering our understanding of nutrient deposition and cycling in these environments. We identified three groups of elements based on their temporal changes. Group 1 elements appear to be cadaver-derived (Na, K, P, S). Group 2 includes three elements that have greater concentrations in soil than would be expected based on cadaver inputs alone, suggesting that these elements partially originate from the soil as a direct result of removal from the exchange (Ca, Mg), or that they are solubilized as a result of soil acidification (Mn). Group 3 includes Al and elements that are found in trace quantities in both humans and soil (Fe, Cu, Zn, Co); all exhibited increases late in the decomposition process, suggesting a gradual solubilization from soil minerals under acidic pH conditions. For forensic applications which seek a reliable marker of the timing of decomposition, elements that change independently of edaphic factors are ideal. Since soil composition, chemistry, and exchange properties vary between regions, it is preferential to consider only elements that are directly cadaver-derived (i.e., group 1 and Se) as potential forensic indicators. Of these cadaver-derived elements, S shows promise as a potential indicator for decomposition progression since its abundances are tied primarily to the breakdown of proteins and larger organic molecules as opposed to participating in the soil exchange. P did not exhibit a specific pattern over our study period, however, its persistence in soil via complexation and gradual deposition resulting from bone diagenesis suggests P may prove to be an indicator of vertebrate decomposition, particularly in the instance of cadaver removal from the decomposition site. Se may also prove useful as a support indicator based upon a broad range of characteristics: microbially

mediated breakdown products of selenoproteins, a similarity of environmental behavior to that of S, and enrichment within plant tissues.

## Supporting information

**S1 Fig. X-ray diffractograms of clay minerals.** Treatments ($K^+$ saturated at 25°C, $K^+$ saturated at 300°C, $K^+$ saturated at 550°C, $Mg^{2+}$ saturated, and $Mg^{2+}$ with glycol saturated) are shown for soils originating from two separate ARF locations (A and B) having no previous decomposition activity. Clay minerals are identified by d-spacing in nm.
(DOCX)

**S1 Table. Elemental concentrations in soil during human decomposition.** Selected elements are those that occur in greatest abundance in the human body and are listed in approximate order of percentage found in the human body from greatest to least. Data are means ± standard deviations for n = 3 replicate donors. Impacted soils that significantly differ from controls based upon Welch T-tests ($p < 0.05$) are presented in bold type. Asterisks indicate levels of significance: $^* p < 0.05$, $^{**} p < 0.01$, $^{***}p < 0.001$.
(DOCX)

**S2 Table. Results of Welch T-tests between decomposition-impacted soils and controls.** Welch T-tests assuming unequal sample variances were conducted at each sampling time point in order to compare differences between impacted soils and controls for pH, electrical conductivity (EC), calcium (Ca), phosphorus (P), potassium (K), sulfur (S), sodium (Na), magnesium (Mg), Iron (Fe), copper (Cu), manganese (Mn), zinc (Zn), selenium (Se), cobalt (Co), boron, (B), and aluminum (Al). P values are shown, and significant differences based upon $p < 0.05$ are presented in bold type. Asterisks indicate levels of significance: $^* p < 0.05$, $^{**} p < 0.01$, $^{***}p < 0.001$.
(DOCX)

**S3 Table. Charge concentrations of exchangeable cations in soil during human decomposition.** Charge concentrations ($\mu mol_c$ $gdw^{-1}$) of exchangeable cations ($Na^+$, $K^+$, $Ca^{2+}$, $Mg^{2+}$) in controls and decomposition-impacted soils are shown for the entire study. Data are means ± standard deviations for n = 3 replicate donors.
(DOCX)

**S4 Table. Trace elemental concentrations in soil during human decomposition.** With the exception of Aluminum (Al), selected elements are those that occur in trace abundance in the human body and are listed in approximate order of percentage found in the human body from greatest to least. Aluminosilicates are common to East Tennessee soils, and Al is included in order to demonstrate increased mobility in soil under decreased pH found in decomposition. Data are means ± standard deviations for n = 3 replicate donors. Impacted soils that significantly differ from controls based upon Welch T-tests ($p < 0.05$) are presented in bold type. Asterisks indicate levels of significance: $^* p < 0.05$, $^{**} p < 0.01$, $^{***}p < 0.001$.
(DOCX)

**S1 File. Raw data for this study.**
(XLSX)

## Acknowledgments

We would like to thank the donors and their families, and the support personnel with the University of Tennessee Forensic Anthropology Center. We also thank Jennifer K. Baer, Mallari

Starrett, Morgan White, and Melanie Stewart for their assistance in laboratory analyses and soil sampling, and Anielle Duncan for assistance at the Anthropology Research Facility. Analytical support for this project was provided by the Water Quality Core Facility at the University of Tennessee, Knoxville, under the management of Adrian Gonzalez, PhD. Finally, we extend our sincerest appreciation to our anonymous reviewers whose feedback has greatly improved this manuscript.

## Author Contributions

**Conceptualization:** Lois S. Taylor, Scott C. Lenaghan, C. Neal Stewart, Amy Z. Mundorff, Dawnie W. Steadman, Jennifer M. DeBruyn.

**Data curation:** Lois S. Taylor.

**Formal analysis:** Lois S. Taylor.

**Funding acquisition:** Scott C. Lenaghan, C. Neal Stewart, Amy Z. Mundorff, Dawnie W. Steadman, Jennifer M. DeBruyn.

**Investigation:** Lois S. Taylor, Adrian Gonzalez.

**Methodology:** Lois S. Taylor, Adrian Gonzalez.

**Project administration:** Scott C. Lenaghan, Jennifer M. DeBruyn.

**Resources:** Scott C. Lenaghan, Jennifer M. DeBruyn.

**Software:** Jennifer M. DeBruyn.

**Supervision:** Jennifer M. DeBruyn.

**Validation:** Lois S. Taylor.

**Visualization:** Lois S. Taylor, Michael E. Essington, Jennifer M. DeBruyn.

**Writing – original draft:** Lois S. Taylor.

**Writing – review & editing:** Lois S. Taylor, Adrian Gonzalez, Michael E. Essington, Scott C. Lenaghan, C. Neal Stewart, Amy Z. Mundorff, Dawnie W. Steadman, Jennifer M. DeBruyn.

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
