## [Decision Letter · Decision Letter 0]

30 Mar 2023

PONE-D-23-02805Soil elemental changes during human decompositionPLOS ONE

Dear Dr. DeBruyn,

Thank you for submitting your manuscript to PLOS ONE. After careful consideration, we feel that it has merit but does not fully meet PLOS ONE’s publication criteria as it currently stands. Therefore, we invite you to submit a revised version of the manuscript that addresses the points raised during the review process. The revisions are minor and we look forward to receiving an updated version.

We look forward to receiving your revised manuscript.

Kind regards,

Victoria Elaine Gibbon

Academic Editor

PLOS ONE

“The authors acknowledge funding by the Defense Advanced Research Projects Agency (Award

No. D20AC00007) to CNS, SCL, AZM, DWS and JMD. The views, opinions, and/or findings expressed are those of the authors and should not be interpreted as representing the official views of policies of the Department of Defense or the United States Government (Approved for Public Release, Distribution Unlimited). Publication fees were granted by the University of Tennessee Librarians Open Publishing Support Fund.”

5. We note that Supplementary Figure S1 in your submission contain [map/satellite] images which may be copyrighted. All PLOS content is published under the Creative Commons Attribution License (CC BY 4.0), which means that the manuscript, images, and Supporting Information files will be freely available online, and any third party is permitted to access, download, copy, distribute, and use these materials in any way, even commercially, with proper attribution. For these reasons, we cannot publish previously copyrighted maps or satellite images created using proprietary data, such as Google software (Google Maps, Street View, and Earth). For more information, see our copyright guidelines: http://journals.plos.org/plosone/s/licenses-and-copyright.

a. You may seek permission from the original copyright holder of Supplementary Figure S1 to publish the content specifically under the CC BY 4.0 license. 

Reviewers' comments:

Reviewer's Responses to Questions

**Comments to the Author**

1. Is the manuscript technically sound, and do the data support the conclusions?

Reviewer #1: Yes

Reviewer #2: Yes

2. Has the statistical analysis been performed appropriately and rigorously? 

Reviewer #1: Yes

Reviewer #2: Yes

3. Have the authors made all data underlying the findings in their manuscript fully available?

Reviewer #1: Yes

Reviewer #2: Yes

4. Is the manuscript presented in an intelligible fashion and written in standard English?

Reviewer #1: Yes

Reviewer #2: Yes

5. Review Comments to the Author

Reviewer #1: PONE-D-23-02805

The authors present a timely and interesting study examining changes in soil elemental composition associated with human decomposition. The manuscript is well written, methods and data analyses seem appropriate, and the authors do not make interpretations beyond the scope of their data. The study is somewhat limited by small sample sizes but considering the nature of the work that is understandable and should not preclude publication. I have a number of minor clarifications but don’t see any reason not to accept this article for publication. However, I am not an expert in the soil physical or chemical analyses performed for this study and as such defer to the opinion of other reviewers or editorial board member.

Minor comments

The control sites were 2m from each body. What was the rationale for this specific distance? Based on the results it sees to be sufficient but no justification is provided in the text.

Table 2. Please clarify what ‘Trial 3’ in the table title refer to.

Figure 3. Branching in the cluster analysis appears to indicate that Mn and Se cluster with Group 1 rather than Group 2. Please outline why these elements are included in Group 1.

Reviewer #2: I highly recommend the presented research for publication. Althoug my positive opinion there are some comments for your consideration before the final publication:

comments to the line 236: The change in mobility under the influence of soil acidity applies to Al, but also to other trace metals. For example, The availability and mobility of Cu and Zn will increase in low pH environments due to the chemical form in which these metals are present in the soil solutions. Maybe it's enough to add this trace metal to the list with the rest, then?

line 401: Szelecz et al. (2018) https://doi.org/10.1016/j.forsciint.2018.02.03 observed a decrease in soil pH (in effect, acidification) under the influence of decomposing Sus scrofa carcasses. A study by Fancher et al. (2017) https://doi.org/10.1016/j.forsciint.2017.08.002 did not conclusively demonstrate that decomposition of human cadavers decreases soil pH. The results were divergent. Barton et al. (2020) https://link.springer.com/article/10.1007/s12024-020-00297-2 in a study compared the decomposition of human and pig cadavers. The effect of cadaver decomposition, regardless of species, affected changes in soil pH levels in a similar way. In my opinion, it is difficult to determine a clear trend here.

line 446: In study of Cholewa et al (2018) DOI: 10.1016/j.forsciint.2022.111208, after about 200 days, the level of inorganic form of phosphorus (P2O5) in the soil with buried pig limbs differed from the control fields, as well as from each other (the depth of buried animal tissues was a differentiating factor).

line 477: Results of Cholewa et al (2018) DOI: 10.1016/j.forsciint.2022.111208 confirm that it was the same with potassium. Plants growing at the site where the pigs' limbs were buried did not have higher levels of K, although there was more of it in the soil - perhaps the plants are not responsible for its loss (as well as for the loss of Na, as demonstrated by the authors of the reviewed paper).

line 517: Was the drop in Ca levels related to the appearance of adipocere? Fiedler et al. (2004) suggested that the calcium was bound in adipocere. Fiedler, Schneckenberger, & Graw, Characterization of Soils Containing Adipocere. Arch Environ Contam Toxicol 47, 561–568 (2004). https://doi.org/10.1007/s00244-004-3237-4

6. PLOS authors have the option to publish the peer review history of their article (what does this mean?). If published, this will include your full peer review and any attached files.

Reviewer #1: No

Reviewer #2: No

---

## [Author Response · Author response to Decision Letter 0]

12 May 2023

Please see uploaded "response to reviewers" file

---

## [Decision Letter · Decision Letter 1]

30 May 2023

Soil elemental changes during human decomposition

PONE-D-23-02805R1

Dear Dr. DeBruyn,

We’re pleased to inform you that your manuscript has been judged scientifically suitable for publication and will be formally accepted for publication once it meets all outstanding technical requirements.

Kind regards,

Victoria E. Gibbon

Academic Editor

PLOS ONE

Additional Editor Comments (optional):

Reviewers' comments:

Reviewer's Responses to Questions

**Comments to the Author**

1. If the authors have adequately addressed your comments raised in a previous round of review and you feel that this manuscript is now acceptable for publication, you may indicate that here to bypass the “Comments to the Author” section, enter your conflict of interest statement in the “Confidential to Editor” section, and submit your "Accept" recommendation.

Reviewer #2: All comments have been addressed

2. Is the manuscript technically sound, and do the data support the conclusions?

Reviewer #2: Yes

3. Has the statistical analysis been performed appropriately and rigorously? 

Reviewer #2: Yes

4. Have the authors made all data underlying the findings in their manuscript fully available?

Reviewer #2: Yes

5. Is the manuscript presented in an intelligible fashion and written in standard English?

Reviewer #2: Yes

6. Review Comments to the Author

Reviewer #2: (No Response)

7. PLOS authors have the option to publish the peer review history of their article (what does this mean?). If published, this will include your full peer review and any attached files.

Reviewer #2: No

---

## [Editor Report · Acceptance letter]

4 Jun 2023

PONE-D-23-02805R1 

Soil elemental changes during human decomposition 

Dear Dr. DeBruyn:

I'm pleased to inform you that your manuscript has been deemed suitable for publication in PLOS ONE. Congratulations! Your manuscript is now with our production department. 

Kind regards, 

on behalf of

Prof Victoria E. Gibbon 

Academic Editor

PLOS ONE